# Thermal, Tensile and Fatigue Behaviors of the PA6, Short Carbon Fiber-Reinforced PA6, and Continuous Glass Fiber-Reinforced PA6 Materials in Fused Filament Fabrication (FFF)

**DOI:** 10.3390/polym15030507

**Published:** 2023-01-18

**Authors:** Mohammad Ahmadifar, Khaled Benfriha, Mohammadali Shirinbayan

**Affiliations:** 1Arts et Metiers Institute of Technology, CNAM, LCPI, HESAM University, F-75013 Paris, France; 2Arts et Metiers Institute of Technology, CNAM, PIMM, HESAM University, F-75013 Paris, France

**Keywords:** material extrusion, rheological behavior, mechanical properties, temperature profile

## Abstract

Utilization of additive manufacturing (AM) is widespread in many industries due to its unique capabilities. These material extrusion methods have been developed extensively for manufacturing polymer and polymer composite materials. The raw material in filament form are liquefied in the liquefier section and are consequently extruded and deposited onto the bed platform. The designed parts are manufactured layer by layer. Therefore, there is a gradient of temperature due to the existence of the cyclic reheating related to each deposited layer by the newer deposited ones. Thus, the stated temperature evolution will have a significant role on the rheological behavior of the materials during this manufacturing process. Furthermore, each processing parameter can affect this cyclic temperature profile. In this study, different processing parameters concerning the manufacturing process of polymer and polymer composite samples have been evaluated according to their cyclic temperature profiles. In addition, the manufactured parts by the additive manufacturing process (the extrusion method) can behave differences compared to the manufactured parts by conventional methods. Accordingly, we attempted to experimentally investigate the rheological behavior of the manufactured parts after the manufacturing process. Thus the three-point bending fatigue and the tensile behavior of the manufactured samples were studied. Accordingly, the effect of the reinforcement existence and its direction and density on the tensile behavior of the manufactured samples were studied. Therefore, this study is helpful for manufacturers and designers to understand the behaviors of the materials during the FFF process and subsequently the behaviors of the manufactured parts as a function of the different processing parameters.

## 1. Introduction

Additive manufacturing (AM) is growing due to the low amount of wasted material in the manufacturing process and the ability to manufacture complex shapes [1,2,3]. There are different techniques concerning the additive manufacturing process, while fused filament fabrication (FFF) is the most commonly utilized technique [4]. Mechanical properties and dimensional accuracy of the FFF-processed parts are affected by the utilized processing parameters during the manufacturing process [5]. The importance and the influence of the assortment of processing parameters have been studied, such as the effect of print speed [6,7], bed platform temperature [8,9,10], liquefier temperature [11], and layer height [12,13]. The evolution of the temperature of the deposited layers is considerably affected by the aforementioned processing parameters during the FFF process [5]. The temperature evolution caused a gradient of the temperature in the structure, which significantly affected the adhesion and the bonding of the deposited layers and consequently the strength of the manufactured parts. Several studies related to the temperature evolution during the FFF process have been conducted [5,14,15].

Christiyan et al. [16] investigated the flexural and tensile strength of the printed ABS composite materials under the different print speed values of 50, 40, and 30 mm/s and the layer height values of 0.3, 0.25, and 0.2 mm. As for the results, it was observed that the lower layer height and print speed values (0.2 mm and 30 mm/s) increased the flexural and tensile strengths. Moreover, several studies have been carried out regarding the FFF process of polymer composite materials [17,18,19]. Durga et al. [20] evaluated the influence of the liquefier temperature and layer height of the deposited layers on the tensile strength of the printed CF-PLA specimens. The manufactured CF-PLA specimens under the lowest selected layer height value and the highest selected liquefier temperature had the higher tensile strength. An investigation was conducted by Ding et al. [21] to determine the influence of liquefier temperature on the mechanical properties of the FFF-processed PEEK and PEI. They discovered that the flexural strength was gradually improved as the temperature increased. Berretta et al. [22] manufactured the reinforced PEEK with 1% and 5% carbon nanotubes (CNTs). They reported that the CNTs did not have a significant effect on the mechanical behaviors of the PEEK-processed specimens. They introduced the nozzle temperature as one of the most crucial parameters in the FFF process, due to its direct contact with the polymer. Yang et al. [23] studied the effect of the thermal processing condition on mechanical behaviors and crystallinity of the PEEK material. Based on the related results, crystallinity increased from 17% to 31% in response to the increase in ambient temperature from 25 to 200 °C.

Few studies have been performed to investigate the effect of the utilized fiber reinforcements on the rheological behavior of the materials during and after the manufacturing process. This study tried to investigate the rheological behavior of the materials during and after the FFF process by considering the role of the fiber reinforcements. The selected materials for this study were PA6, short carbon fiber-reinforced PA6 (CF-PA6), and continuous glass fiber-reinforced CF-PA6 composite materials.

## 2. Materials and Methods

### 2.1. Materials

The selected materials for this study were polyamide 6 (PA6) and short carbon fiber-reinforced polyamide 6 (Onyx or CF-PA6) produced by MarkForged^®^. The chopped carbon fibers had a mass content of 6.5% in the CF-PA6 filament based on the pyrolysis process. The characteristics of the utilized filaments as the raw materials are presented in Table 1.

As for the investigation of the fiber-reinforcement impact and the processing parameter effects on the rheological behavior of the materials during the FFF process, a single wall layer specimen (Figure 1) was designed. This specimen let us study the effect of the selected processing parameters on the adhesion and the bonding of the deposited layers. In addition, the location of the required specimens for the subsequent characterizations are determined in Figure 1. Two different printers were utilized during our studies. To study the effects of the processing parameters, Flashforge ADVENTURER-3 (from China) was utilized. Moreover, as for studying the infill percentage, infill pattern effects, and fatigue behaviors, a Markforged-Mark Two (from the USA) printer was utilized. This is because Markforged-Mark Two printers can provide the possibility of manufacturing composite objects with continuous reinforcement.

### 2.2. Methods

#### 2.2.1. In Situ Monitoring of Temperature Evolution

An Optris PI450 infrared camera was applied in the conducted study of the processing parameter impacts on the thermal and mechanical properties of the polymer and polymer-based composites using the FFF process. The stated infrared camera was positioned at a specific predetermined distance from the extrusion location of the printer. This attempt aimed to monitor and observe a consistent plain field of view (FOV) across all consecutive layers. The temperature rise during the performed fatigue test was monitored and measured by the stated infrared camera as well. Regarding some technical specifications of the infrared camera used in this experiment, frequency, accuracy values, wavelength range, optical resolution, and the rate of the frames were 32 Hz, 2%, 8–14 μm, 382 × 288 pixels, and 80 Hz, respectively. 

#### 2.2.2. Microstructural Observations

Using a scanning electronic microscope (HITACHI 4800 SEM, manufactured in Japan), observations and image analyses were performed to qualitatively analyze the composite microstructure, especially in relation to damage assessment. In this study, an optical microscope (OLYMPUS BH2, manufactured in Japan) was used to assess the quality of the manufactured samples at the various selected processing parameters with magnifications from 100 to 500 mm.

#### 2.2.3. Differential Scanning Calorimetric (DSC)

By means of differential scanning calorimetry (Q1000), both raw filament materials and printed specimens could be assessed for their respective glass transition (T_g_), crystallization temperatures (T_c_), and their heat capacities based on the selected processing parameters. DSC characterization was performed on the raw filament materials (CF-PA6 and PA6) over three temperature ramps: range of 20 to 220 °C under a rate of 10 °C/min. This attempt caused an elimination of the thermal history of the filaments concerning their production process. Furthermore, the printed and fabricated specimens were analyzed by DSC in two ramps (heating and cooling).

#### 2.2.4. Thermo-Mechanical Behavior Analysis (DMTA)

Under the multi-frequency condition, DMTA flexural tests were performed on the printed samples using the DMA Q800 instrument from the TA Company, in order to determine the major transition temperatures and viscoelastic characteristics. This characterization was utilized to study the viscoelastic behavior of the material during the fatigue test and the subsequent temperature rise. DMTA characterization was performed in the temperature range of 30 to 80 °C, frequencies of 1, 2, 5, 10, and 30 Hz, and temperature rate of 2 °C/min.

#### 2.2.5. Quasi-Static Tensile Test

As was stated, two different conditions for studying the tensile behavior were conducted. The first condition was considered to study the effect of the processing parameters on the manufacturing of the polymer and composite specimens (Figure 1). Therefore, based on ISO 527-2, tensile test specimens were sliced/cut from the printed single-wall layer sample (Figure 1), in order to conduct the related study of the first condition. The required tensile specimens were cut from the printed single-wall samples utilizing a tensile sample-cutting die and applying the homogenous force. In addition, the homogeneity of the printed single-wall layers was ensured by using the caliper for thickness measurement. Following the cutting of the tensile test specimens from the single-wall layers, a caliper was used to ensure that gauge length dimensions were uniform. In addition, the observation of the samples under the optical microscope (OM) to control the quality of the specimens after the manufacturing and cutting process was taken into account. The second condition was applied to study the impact of fill percentage of the polymer and the different determined densities and directions of the utilized continuous reinforcements. Moreover, in order to investigate the rheological behavior of the additive manufactured specimens after the production process, the related tensile test specimens were printed based on the standard ISO 527-1. The quasi-static tensile experiment was conducted by means of the INSTRON 5966 machine with a displacement rate of 5 mm/min and a loading cell of 10 kN. A minimum of three specimens were prepared for each condition in order to conduct the quasi-static tensile tests.

#### 2.2.6. Three Points Bending Fatigue Tests

As for the fatigue test of the fabricated specimens concerning the study of the rheological behavior of the FFF-processed composite materials (after the manufacturing process), a three-points bending fatigue test was conducted at various applied maximum strains (ε_max_). The considered strain ratio was 0.1 (R_σ_ = 0.1); moreover, the related mean strain level was 0.55 ε_max_. This test was conducted on the short carbon fiber-reinforced polyamide 6 (Onyx or CF-PA6) and the continuous glass fiber-reinforced polyamide 6 (Onyx + GF) composite specimens to be able to study the effect of the continuous glass fiber reinforcement and its subsequent impact on the rheological behavior of the FFF-processed specimens. During the three-point bending fatigue test, the temperature of the specimens was raised, which was measured by the infrared camera to determine the importance and influence of the utilized continuous glass fiber reinforcement in this phenomenon. The dimensions of the prepared rectangular composite specimens were 120 × 10 × 4 mm^3^. In order to perform a fatigue test, at least three specimens were prepared for each condition.

## 3. Results and Discussions

### 3.1. Study of the Rheological Behavior of the Polymer and Polymer Composite Materials during the FFF Process

The rheological behavior of the polymer and polymer composite materials during the FFF process are investigated in this section. In addition, the modifications in the rheological behavior of the materials as the consequence of the short carbon fiber existence are discussed. As part of this effort, four main processing parameters were considered to investigate their impacts on the tensile behavior of the manufactured specimens. In addition, the in situ temperature evolution during the manufacturing process was monitored. The selected processing parameters were print speed (13, 15, and 17 mm/s), liquefier temperature (220, 230, and 240 °C), layer height (0.1, 0.2, and 0.3 mm), and bed platform temperature (25, 50, 60, and 80 °C). The aforementioned processing parameters were studied individually by considering the rest of the processing parameters as constant. The reference values concerning each of the selected parameters were 15 mm/s as print speed, 240 °C as liquefier temperature, 0.1 mm as layer height, and 25 °C as bed temperature. The selected materials for this section were PA6 and CF-PA6 [5].

#### 3.1.1. Effect of the Print Speed

In light of the importance of the print speed parameter in production time, it was studied to how it impacts the rheological behaviors. To study the impact of this manufacturing processing parameter, 13, 15, and 17 mm/s as the different print speed values were taken into consideration. According to Figure 2, the tensile strength and the crystallinity percentage pertaining to the manufactured specimens with the above-selected print speed values were evaluated. The tensile strength values of 57.78 ± 0.84, 68.92 ± 0.9, and 69 ± 0.05 MPa concerning the manufactured PA6 specimens and 65 ± 0.5, 55 ± 0.6, and 63 ± 1.3 MPa concerning the manufactured CF-PA6 specimens pertaining to the selected print speed values of 13, 15, and 17 mm/s were obtained, respectively.

Figure 3 depicts the temperature profile, obtained from the in situ monitoring of the temperature evolution concerning the PA6 and CF-PA6 specimens produced by the above-stated print speed values. As printing speed was increased, the measured temperature of the first printed layer stayed above the related crystallization temperature values in all examined PA6 and CF-PA6 specimens. The print speed affected the cooling time and polymer arrangement and the resultant crystallinity degree. All selected print speeds yielded higher crystallinity degree values for CF-PA6 specimens than PA6 specimens.

#### 3.1.2. Effect of the Liquefier Temperature

The impact of the liquefier temperature on the rheological behavior of the PA6 and CF-PA6 specimens were investigated. The decided liquefier temperature values for this study were 240, 230, and 220 °C. Figure 4 depicts the crystallinity percentage and tensile strength concerning the manufactured specimens with the above-selected liquefier temperature values. The tensile strength values of 55.48 ± 0.78 and 68.92 ± 0.9 MPa concerning the manufactured PA6 specimens by the liquefier temperature values of 230 and 240 °C were achieved, respectively. One can note that the lowest strength was observed in the printed specimens by the liquefier temperature values of 220 °C. This liquefier temperature value could not provide a suitable fluidity of the material during the FFF process, which caused low and inappropriate adhesion between the deposited layers. This low adhesion was revealed during the punching/slicing process of the printed single wall samples for preparation of the required tensile test specimens by a fracture that occurred in the interface of the deposited layers. This undesirable fracture did not allow us to have tensile test specimens to evaluate the tensile strength behavior of the manufactured PA6 specimens by the liquefier temperature value of 220 °C. In addition, the tensile strength values of 49 ± 1.5, 51 ± 3, and 55 ± 0.6 MPa concerning the manufactured CF-PA6 specimens pertaining to the selected liquefier temperature values of 220, 230, and 240 °C were obtained, respectively. The determined crystallinity degree of the PA6 samples printed at the liquefier temperatures of 220, 230, and 240 °C were 12.51%, 12.75%, and 14.40%, respectively, while the obtained crystallinity degree of the CF-PA6 samples printed at the liquefier temperatures of 220, 230, and 240 °C were 19.97%, 20.26%, and 20.51%, respectively.

Regarding the effect of the short fiber reinforcement and the consequent differences between the FFF-processed polymer and polymer composite material, firstly, the crystallinity percentage values were highlighted. Based on the obtained results from the DSC characterization of the FFF-processed CF-PA6 and PA6 samples, the crystallinity percentage values of the CF-PA6 specimens were higher than the PA6 specimens manufactured under the same processing parameters. The chopped carbon fibers in CF-PA6 were the superior sites and spots for the nucleation of the crystalline sections. Furthermore, another difference between the FFF-processed polymers and polymer composite materials is their tensile strength. Regarding compression between the obtained tensile strength values of CF-PA6 and PA6, it was understood that the single-wall FFF-manufactured CF-PA6 specimens exhibited lower strength. The short carbon fibers as the solid component in the molten polymer decreased the fluidity of the CF-PA6 during the FFF process compared with PA6 material. The stated decrease in the fluidity during the manufacturing process can be illustrated by the narrow width zones in the manufactured specimens (Figure 5). The stated zones are the preferable rupture sections because of their stress concentration feature in the structure.

#### 3.1.3. Effect of the Layer Height

CF-PA6 and PA6 specimens were investigated for their rheological behavior influenced by the layer height. The investigation of the effect of the short fiber reinforcement on the rheological behavior of the materials was carried out by selecting the three different layer height values of 0.1, 0.2, and 0.3 mm, during and after the FFF process. Figure 6 represents the tensile strength values related to the manufactured specimens by the different selected layer height values.

The tensile strength values of 55 ± 0.6, 49 ± 2.5, and 56 ± 2.5 MPa concerning the manufactured CF-PA6 specimens by the layer height values of 0.1, 0.2, and 0.3 mm were achieved, respectively. For better investigation of the effects of the layer height and the short carbon fiber reinforcements on the rheological behavior of the material during the FFF process, a thermal camera for the in situ temperature monitoring was utilized. By considering the obtained results from the tensile test and the in situ temperature monitoring during the FFF process and their subsequent correlations, the existence of the two effective phenomena was declared. The stated two phenomena were firstly the fluidity decrease in the printed materials and secondly the residual/retained temperature increase in the deposited layers during the cyclic raised temperature during the FFF process. The peaks (upper limits) of the obtained time–temperature curves during the manufacturing process revealed the first-stated phenomenon, and the lower limits of these curves revealed the second-stated phenomenon. By increasing the layer height during the manufacturing process from 0.1 to 0.2 mm, the first phenomenon was more highlighted and effective. The decreased fluidity was found according to the decrease in the measured temperature profile peaks by increasing the selected layer height from 0.1 to 0.2 mm (Figure 7a). This decrease in the fluidity of the deposited layers during the FFF process of CF-PA6 specimens resulted in a lower tensile strength of the manufactured specimens with a layer height of 0.2 mm compared to the printed specimens with a layer height of 0.1 mm. As for determining the effect of the short carbon fibers, the obtained tensile strength from printed CF-PA6-specimens and PA6 specimens in the same layer height values (0.1 and 0.2 mm) were compared. The tensile strength of PA6 specimens with layer heights of 0.1 mm and 0.2 mm were 68.92 ± 0.9 and 72.57 ± 0.8 MPa, respectively. This subsequent increase in the tensile strength with the increase in layer height demonstrated that the decreased fluidity phenomenon did not predominate in the PA6 specimens. The short carbon fibers could also be effective on the stated decreased fluidity. As further explain, the short carbon fiber reinforcements of the solid components hindered the fluidity of the polymer (matrix) as part of the liquid component during the FFF process. However, the second phenomenon, which was a residual/retained temperature increase in the deposited layers, was dominant during the FFF process of the CF-PA6 specimens, with a layer height of 0.3 mm. The residual/retained temperature increase in the deposited layers was found based on the bottom of the measured temperature profile (Figure 7b). This phenomenon during the FFF process of the CF-PA6 specimens resulted in a higher crystallinity percentage and tensile strength of the specimens with a layer height of 0.3 mm.

#### 3.1.4. Effect of the Bed Temperature

As for the study of the influence of the last selected processing parameter, the importance of the bed platform temperature on the manufactured specimens was evaluated. The studied materials were PA6 and CF-PA6. Four different bed temperatures of 80, 60, 50, and 25 °C were taken into account. According to the monitoring of the temperature during the process, upon deposition of the first layer of the materials (PA6 and CF-PA6), the monitored temperature values ascended to the values of crystallization temperature (Figure 8).

The dimensional stability/accuracy of the printed specimens was evaluated by means of optical microscopy. Based on the performed visual comparisons, the manufactured parts under a bed temperature of 80 °C had less dimensional accuracy due to the observed inconsistency and dissonance concerning the thickness of the deposited PA6 layers (Figure 9). The stated inconsistency was observed in the FFF-processed specimens with a bed temperature of 80 °C.

### 3.2. Study of the Rheological Behavior of Polymer and Polymer Composite Material after FFF Process

Investigations regarding the impact of some important FFF processing parameters of the polymer-based composites on the inter-layer adhesion (bonding) of the deposited filaments were performed. It was proven that temperature profile evaluation of the printed layers has a significant effect on the bonding of adjacent filaments. Failure stress/strain can be the indicators to determine the mechanical properties of FFF-manufactured products. In this section, using optimized processing parameters (liquefier temperature: 240 °C, print speed: 15 mm/s, layer height: 0.1 mm, platform temperature: 25 °C), the mechanical properties of the fabricated composite material under monotonic and fatigue loadings were analyzed (Figure 10). As part of the monotonic loading tests, the effect of fill pattern and fill density of the polymer matrix component as well as the density and direction of the continuous reinforcement were studied. Moreover, the effects of continuous reinforcement on the fatigue behavior of the FFF-processed specimens were investigated [24].

#### 3.2.1. Quasi-Static Tensile Property

##### Influence of Fill Patterns

Using the Mark Two printer, the tensile specimens were printed into three fill patterns: triangular, rectangular, and hexagonal. Comparisons were made based on the tensile strength of the manufactured samples made of CF-PA6. Therefore, the obtained tensile strength values of the samples concerning the stated fill pattern were compared to each other as well as to the solid fill pattern (whose infill percentage was 100%) (Figure 11).

The obtained tensile strength of the manufactured CF-PA6 specimens in the triangular, rectangular, hexagonal fill patterns, and the solid fill were 18.79 ± 1.19, 19.84 ± 1.66, 19.99 ± 1.32, and 30.31 ± 5.5 MPa, respectively (Figure 11). The tensile strength of the printed samples in the rectangular and hexagonal fill patterns were about 5.9% and 6.4% higher than the printed samples in the triangular fill pattern, respectively. The tensile strength of the solid fill-printed samples was about 61.3% higher than the rectangular fill pattern-printed samples. In other words, the tensile strengths of the manufactured specimens in the triangular, rectangular, and hexagonal fill patterns were close to each other. However, the highest tensile strength was related to the solid fill samples. As it is observed from Figure 12, the fracture surface of the solid filled-printed samples was more brittle and more homogenous, in comparison with the other samples.

##### Effect of the Continuous Reinforcement on Different Fill Patterns

CF-PA6 filaments with continuous glass fiber were utilized to manufacture the specimens in the stated fill patterns: triangular, rectangular, and hexagonal, as well as in the solid fill pattern (whose fill percentage was 100%). Using CF-PA6 reinforced with continuous glass fiber, the obtained tensile strengths concerning the hexagonal, rectangular, triangular fill patterns, and the solid fill specimens were 60.04 ± 4.16, 66.7 ± 2.63, 69.6 ± 3.05, and 78.35 ± 0.87 MPa, respectively. Therefore, the highest tensile strength is related to the solid fill samples. In addition, by shifting the fill pattern of the printing process of the samples, which were made of CF-PA6 reinforced with continuous glass fibers in the hexagonal infill pattern (the weakest obtained strength) and the solid fill, the tensile strength improved almost 30.5% (half of the situation in which no continuous glass fiber was used). The highest tensile strength was related to the solid fill samples (Figure 13).

Figure 14 illustrates the effect of using continuous glass fiber on the tensile strength of the FFF-processed specimens in the solid fill pattern. By utilizing the continuous glass fiber, the strength of the solid fill samples increased significantly by about 158.5%.

#### Effect of the Density of Continuous Reinforcement

Regarding the effect of the density of the continuous reinforcement layers, a rectangular sample with thickness of 3.5 mm was designed as the tensile test specimens (Figure 15). The continuous fiberglass was considered for manufacturing the PA6 reinforced with continuous glass fiber composite samples. Thus, the tensile test specimens were PA6 composites reinforced with continuous glass fibers. The 0.1 mm value was selected as the layer height of the deposited PA6 layers. In addition, the layer height of the printed fiber glass was 0.1 mm. Thus, the samples were manufactured by depositing 35 layers during the layer-by-layer fabrication process. As for studying the effect of the density of the continuous glass fiber layers, different quantities of the printed continuous glass fiber layers were considered to manufacture the related samples to compare their tensile strengths. The different quantities of the continuous glass fiber layers of 2, 4, 6, 8, and 10 out of the total printed layers (35 layers) were applied in the manufactured specimens. The related tensile strengths of the specimens made up of 2, 4, 6, 8, and 10 layers were 36.38 ± 0.53, 144.26 ± 3.18, 187.55 ± 3.35, 226.34 ± 5.4, and 233.72 ± 4.32 MPa, respectively. The tensile strength of the non-reinforced samples, which were processed by 35 layers made of PA6 layers without any continuous glass fiber, was 34.63 ± 2.5 (Figure 16).

As was found, by increasing the quantities of the continuous glass fiber layers, the tensile strengths of the manufactured samples were increased. The tensile strengths significantly increased by enhancing the quantity of the continuous glass fiber layers from two to eight. However, the tensile strength increments of the manufactured samples from eight to ten reinforced layers were less than the related increases in strength provided by enhancing the quantities of the continuous glass fiber layers from two to four. These obtained results are correlated to the weaker adhesion strength between the glass fiber reinforcement layers and the polymer matrix layers in comparison with the polymer-to-polymer layers. By enhancing quantity of the reinforcement layers from two to eight, the tensile strengths of the samples were increased because of the existence of the continuous reinforcement, which resists against the applied tensile stress. The nature of the weaker interface strength of the glass fiber reinforcement layers and the polymer matrix layers in the manufactured samples, which consists of more than eight reinforcement layers, dominated the impact of the existing reinforcement. Thus, we can consider that in our manufactured samples, the fabricated specimens with eight layers showed the optimal density in the continuous reinforcement.

##### Effects of the Continuous Reinforcement Direction

According to the recently stated results, concerning the performed study on the effect of the density of the continuous reinforcement, it was concluded that there is an optimal quantity layer for the continuous reinforcement layers in the manufactured reinforced polymer composites via the FFF process. In the case of our rectangular tensile test sample, eight layers of the continuous glass fiber layers prepared an optimal tensile behavior.

Therefore, the continuous glass fiber-reinforced PA6 composite samples, reinforced by eight layers, were considered for studying the tensile behavior of the manufactured samples with different continuous reinforcement directions. As for studying the effect of the continuous reinforcement direction, five direction values of 0°, 30°, 45°, 60°, and 90° were considered for continuous glass fibers. The tensile strength values of the related manufactured samples with continuous directions of 0°, 30°, 45°, 60°, and 90° were 226.34 ± 5.4, 46.8 ± 2.05, 37 ± 1.1, 36.81 ± 1.1, and 36.6 ± 0.79 MPa, respectively (Figure 17).

As is observed, the tensile strength of the manufactured samples with continuous reinforcement in the 0° direction had the highest strength in comparison with the other fabricated samples under the different reinforcement directions. As the direction of the continuous reinforcement increased from 0° to 90°, the tensile strength decreased. A significant drop in tensile strength was observed by the change in continuous reinforcement direction from 0° to 30°. The tensile strengths of the manufactured samples with 45°, 60°, and 90° as the reinforcement directions were close to each other. In addition, the tensile strength of the fabricated samples with a continuous reinforcement direction of 90° was close to the strength of the unreinforced samples (33.65 ± 2.8 MPa). Thus, it was found that the manufactured samples with a reinforcement direction of 0° had the highest tensile strength, while as the direction of reinforcement decreased, the tensile strength decreased as well. In the case of the samples with a reinforcement direction of 0°, the glass fibers resisted against the tensile stress well because it was in the direction of the applied strength. The fabricated samples with the reinforcement direction of 90° had the weakest strength because of the perpendicular directions of the applied stress and reinforcement. Thus, in the case of the samples with the glass fiber direction of 90°, the impact of the existence of the reinforcement on the tensile strength was not so manifest and sensible. The close tensile strength value of the specimens with a reinforcement direction of 90° to the unreinforced specimens can be evidence of this phenomenon. The macroscopic fracture observations exhibited the same fracture direction as the continuous reinforcement direction (Figure 18).

#### 3.2.2. Assessment of Fatigue Property

##### Influence of Continuous Reinforcement on Fatigue Behavior

Wöhler curves for CF-PA6 and CF-PA6 reinforced with continuous glass fiber samples are shown in Figure 19. Derived from three points, bending fatigue tests were conducted at a frequency of 10 Hz. CF-PA6 reinforced with continuous glass fiber specimens had a fatigue life of about 8000 cycles, while CF-PA6 specimens had a fatigue life of about 200,000 cycles in the applied strain equal to 4.5%. The obtained Wöhler curves of the CF-PA6 reinforced with continuous glass fiber specimens and the CF-PA6 specimens from the conducted fatigue test at a frequency of 10 Hz were of linear and bi-linear forms, respectively. It was found that there was a small difference between the obtained Wöhler curves of the CF-PA6 specimens and the CF-PA6 reinforced with continuous glass fiber specimens at high amplitude, while the related Wöhler curves significantly shifted at low strain amplitudes (Figure 19).

##### Young’s Modulus Evolution and Self-Heating Phenomenon and Relative Young’s Modulus Evolution

Figure 20 illustrates the evolution of the relative stress through the conducted fatigue testing of continuous glass fiber-reinforced CF-PA6 and CF-PA6 samples at the determined strain amplitudes on the curves. For CF-PA6 and CF-PA6 reinforced with continuous glass fiber samples, mechanical fatigue (MF) mostly dominated the fatigue behavior due to damage phenomenon, while for high amplitude and low cycles, thermal fatigue (ITF) dominated the fatigue behavior of composite specimens.

Based on their plots, it is evident that for high loading amplitudes, the dynamic modulus decreases rapidly in a linear regime of the logarithmic curve until the specimen fails. The dynamic modulus of applied low amplitudes deviates into three decreasing regimes–a swift one during the initial cycles (I), a gradual one (II), and a drastic decline (III) just before fracture. In addition, it was found that there is an upsurge in damage kinetics in the case of CF-PA6 reinforced with continuous glass fibers specimens compared to CF-PA6 specimens. This can be due to the existence of the weak polymer–continuous reinforcement interfaces in these specimens (Figure 21).

Variation in the amplitude of the applied loading results in self-heating for CF-PA6 and continuous glass fiber-reinforced CF-PA6 during the fatigue test. This phenomenon affects the viscosity of the polymer as a function of temperature. Figure 22 depicts the temperature rise during the fatigue test concerning the utilized frequency of 10 Hz.

It was observed that the induced temperature of CF-PA6 increased up to about 60 °C, which is in the glass transition zone. Consequently, the stiffness of the matrix agent (polymer) decreased based on the obtained loss factor evolution versus the temperature curve from the performed DMTA test, as shown in Figure 23.

Therefore, in addition to the development of damage, the fatigue failure of the CF-PA6 specimens was caused by the viscous behavior evolution of the polymer agent and the consequent brittle–ductile transition. It is vital to consider the induced temperature during the fatigue. The maximum induced temperature during the fatigue cycles is a crucial point for considering prognosticating the fatigue behavior of the printed polymer and the continuous fiber-reinforced polymer composite materials. The maximum induced temperatures during the fatigue test of the 3D-printed CF-PA6 and reinforced CF-PA6 with continuous glass fiber samples were measured, which are illustrated in Figure 22. According to this figure, the importance of the continuous fiber reinforcement in the fatigue behavior of the reinforced CF-PA6 with continuous glass fiber material is obvious. The measured induced temperature values related to the performed strains concerning the reinforced CF-PA6 with continuous glass fiber specimens were lower than in the CF-PA6 (without continuous reinforcement) specimens. In addition, the slopes of the curves concerning both composite materials, which is representative of the maximum induced temperature rate just before failure, are the same. Both mechanical fatigue (MF) and induced thermal fatigue (ITF) were observed. As shown in Figure 22, regarding the glass transition temperature zone of the utilized material, the commencement of the ITF-dominated zone can be determined.

##### Fatigue Fractography Analysis

Figure 24 depicts the fracture microscopic observations concerning reinforced CF-PA6 with continuous glass fiber and CF-PA6 specimens for studying the influence of the applied loading amplitude on fracture behavior. According to the conducted fractography, the fracture surface of CF-PA6 with continuous glass fiber specimens consisted of continuous glass fiber pull-out and subsequent debonding of the deposited layers (Figure 24a). The fracture surface of CF-PA6 revealed the debonding of the deposited layers and the subsequent crack propagation inside the deposited layers (Figure 24b).

## 4. Conclusions and Perspectives

The primary purpose of this research was to study the rheological characteristics of the materials during and after the FFF process. As a result of this study, we can gain a better understanding of the behaviors of polymers and polymer composite materials, as well as how reinforcements function during and after the FFF process. The obtained results will enable us to improve the FFF-processed parts. The fundamental problem of the produced parts by the FFF process is the presence of the porosities/voids inside the manufactured parts and the subsequent poor adhesion between the deposited layers. In this study, influences of the various processing parameters on the material behaviors during and after the FFF process were investigated. The physicochemical, thermal, and mechanical assessments exhibited that the crystallinity degree was influenced by the processing parameters modification. The crystallinity degree modifications could affect material diffusion during the cooling stage and the subsequent bonding of the deposited adjacent filaments. The strength of this bonding directly affects the mechanical behaviors of the manufactured specimens. Therefore, a precise and local measurement of the temperature on the scale of the diameter of the filaments during the FFF process was performed. In addition, by comparing the obtained temperature evolution curves from PA6 and CF-PA6 specimens, it was found that the existence of the chopped carbon fibers could affect the temperature evolution curves during the FFF process.

The obtained results from the three-point bending fatigue and the tensile behavior of the manufactured composite materials reveal:-The FFF-processed CF-PA6 specimens with solid fill patterns had superior mechanical properties and stiffness under tension.-Based on the study of the effect of the continuous glass fiber reinforcements on the fatigue behavior of the manufactured specimens, a reduction in fatigue life was observed in the manufactured CF-PA6 and continuous glass fiber-reinforced CF-PA6 specimens due to an increase in the induced temperature. By considering this induced temperature, known as self-heating, we can determine how a polymer behaves viscously, especially at the glass transition zone. It was observed that CF-PA6 reinforced with continuous glass fiber specimens exhibited a lower level of self-heating compared to CF-PA6 specimens without continuous glass fiber reinforcement during the fatigue test. Based on the microscopic analysis of the damage mechanisms during the performed fatigue tests, the fracture surface of the continuous glass fiber-reinforced CF-PA6 specimens was characterized by continuous glass fiber pull-out and subsequent debonding of the deposited layers.-As for the study of the density of continuous glass fiber in the FFF process of polymer-based composites, by increasing the density of the reinforcement layers, the strengths of the manufactured specimens were increased until a specific density value, and then, they became stable.-As for studying the effect of the direction of the continuous reinforcement in the FFF-processed polymer composite materials, the five directions of 0°, 30°, 45°, 60°, and 90° were investigated. The highest strength was related to the manufactured parts, in which the reinforcements were in the applied stress direction (0°). As the direction of the utilized reinforcement deviated more from the applied stress direction, the strength of the manufactured parts decreased. The fracture of the tensile-tested samples occurred in the direction of the deposited continuous reinforcement.

In the next study, the effect of other continuous reinforcements such as carbon fiber and Kevlar fiber on the tensile and the fatigue behaviors of the printed specimens will be studied. Therefore, their effects will be compared with each other.

## Figures and Tables

**Figure 1 polymers-15-00507-f001:**
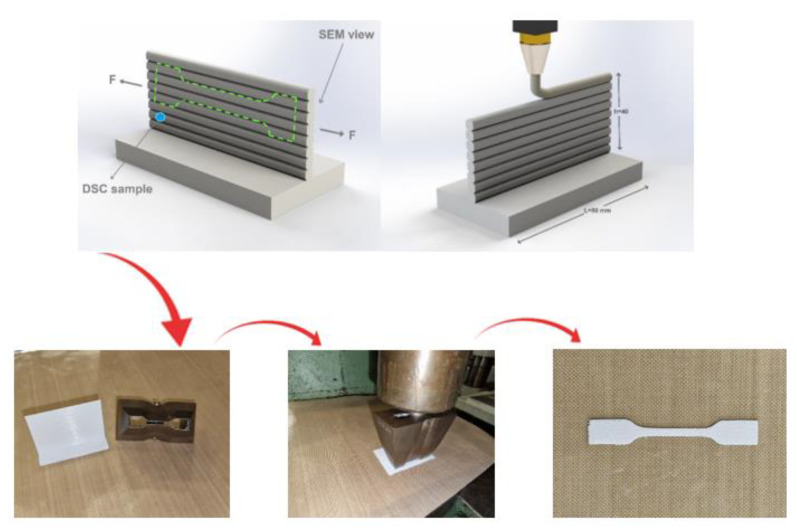
The printed single-wall specimens and preparation of the tensile test specimens [5].

**Figure 2 polymers-15-00507-f002:**
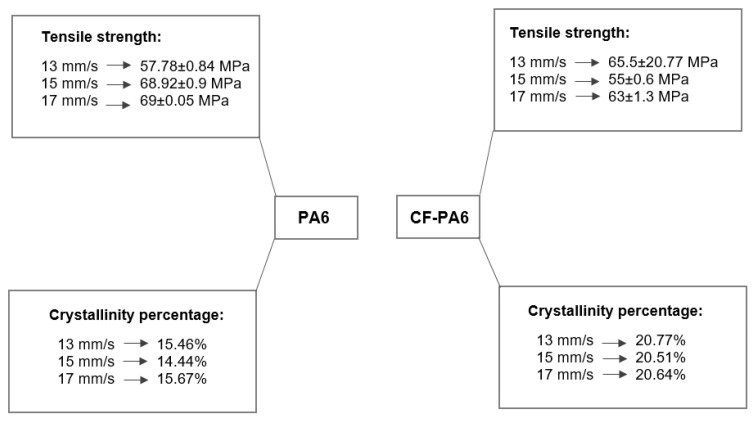
Obtained tensile strength and crystallinity percentage values concerning the print speed effect investigation.

**Figure 3 polymers-15-00507-f003:**
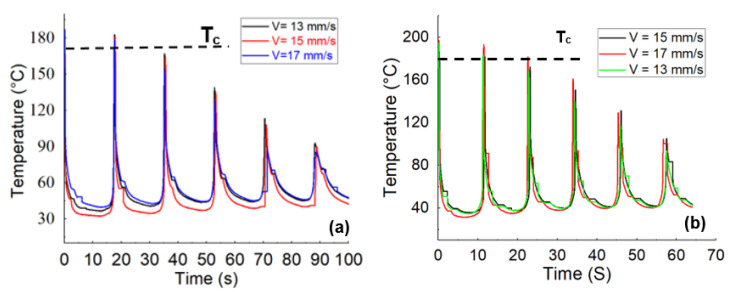
The obtained temperature evolution from the in situ temperature monitoring (**a**) PA6 and (**b**) CF-PA6 [5] under different print speeds.

**Figure 4 polymers-15-00507-f004:**
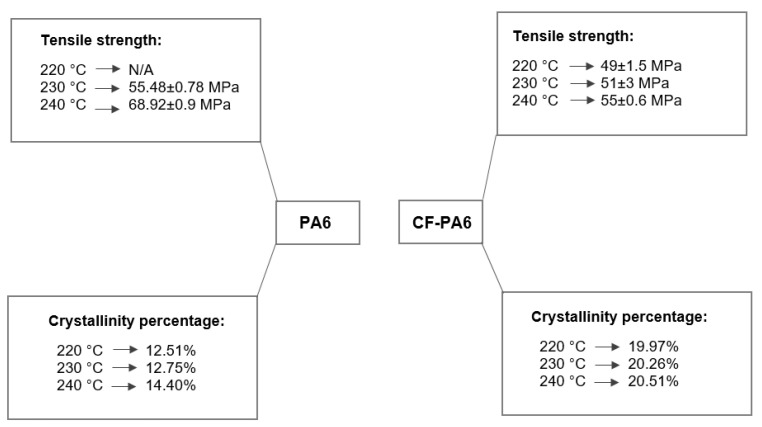
Obtained tensile strength and crystallinity percentage values concerning the liquefier temperature effect investigation.

**Figure 5 polymers-15-00507-f005:**
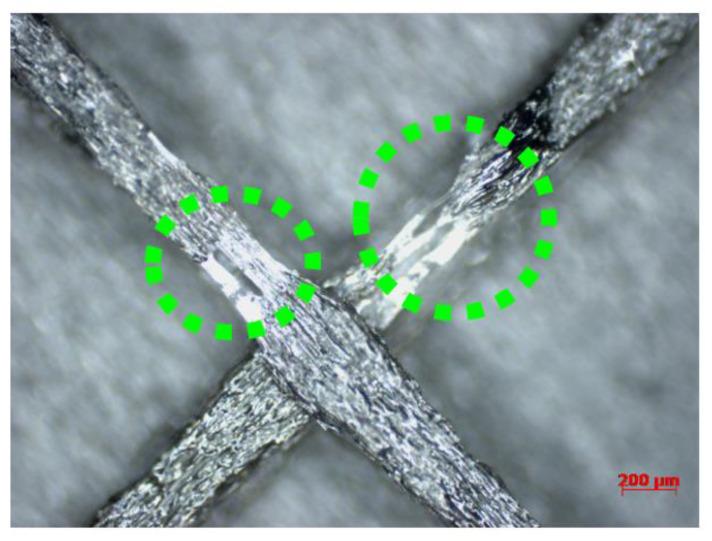
Narrow zone of deposited CF-PA6 filament.

**Figure 6 polymers-15-00507-f006:**
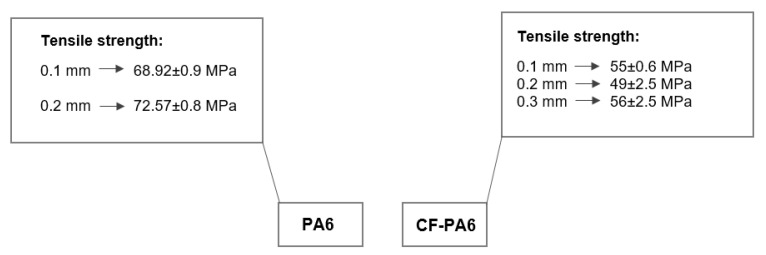
Obtained tensile strength values concerning the layer height effect investigation.

**Figure 7 polymers-15-00507-f007:**
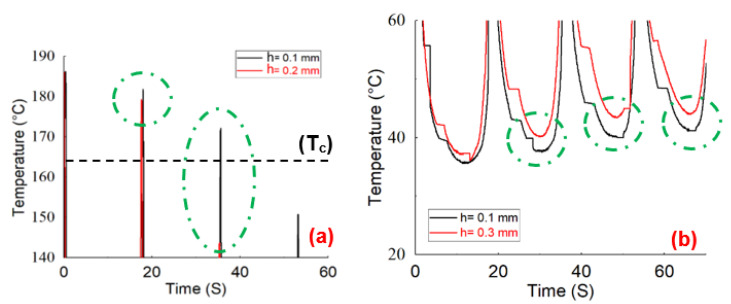
Monitoring of the temperature rise during the additive manufacturing process of CF-PA6 specimens with the different selected layer height values: (**a**) upper limits and (**b**) lower limits.

**Figure 8 polymers-15-00507-f008:**
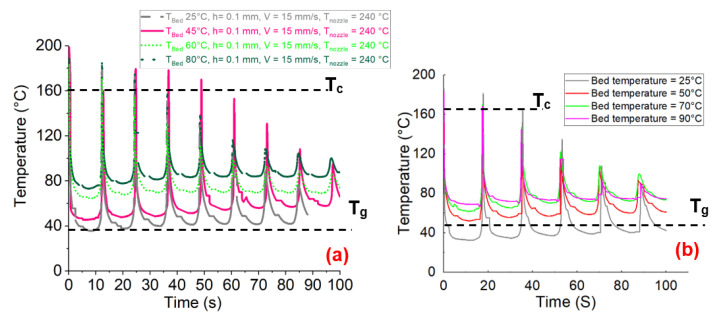
Monitoring of the temperature evolution during the FFF process concerning the different selected bed temperature values: (**a**) CF-PA6, (**b**) PA6.

**Figure 9 polymers-15-00507-f009:**
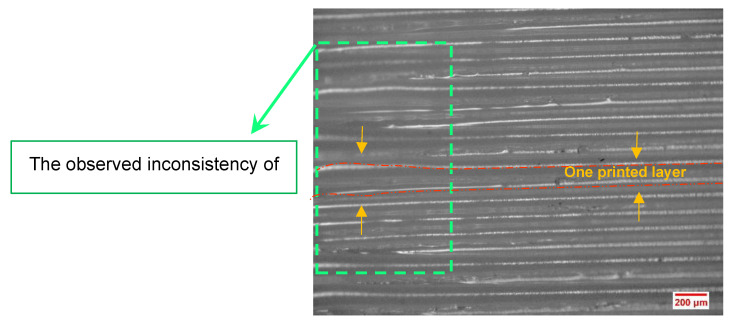
Optical microscopy of the printed PA6 specimen under a bed temperature value of 80 °C.

**Figure 10 polymers-15-00507-f010:**
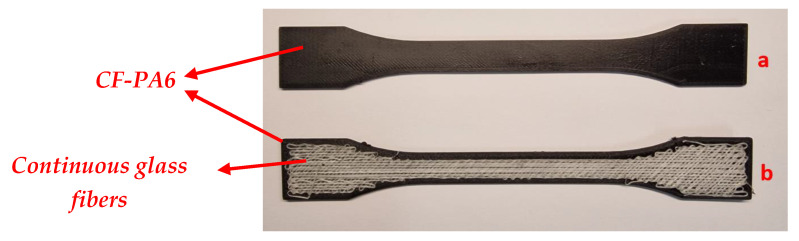
The manufactured CF-PA6 (**a**) and CF-PA6 reinforced with continuous glass fiber (**b**) specimens.

**Figure 11 polymers-15-00507-f011:**
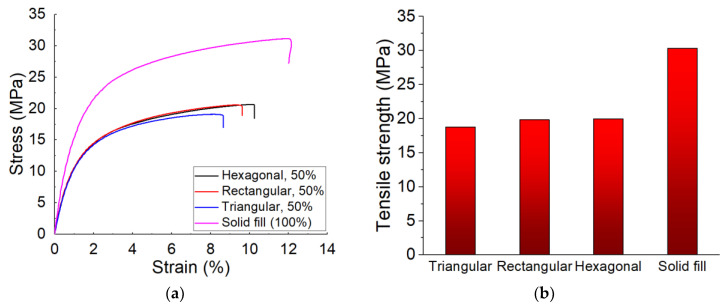
Obtained tensile test curves (**a**) and obtained strength (**b**) of CF-PA6 specimens [24].

**Figure 12 polymers-15-00507-f012:**
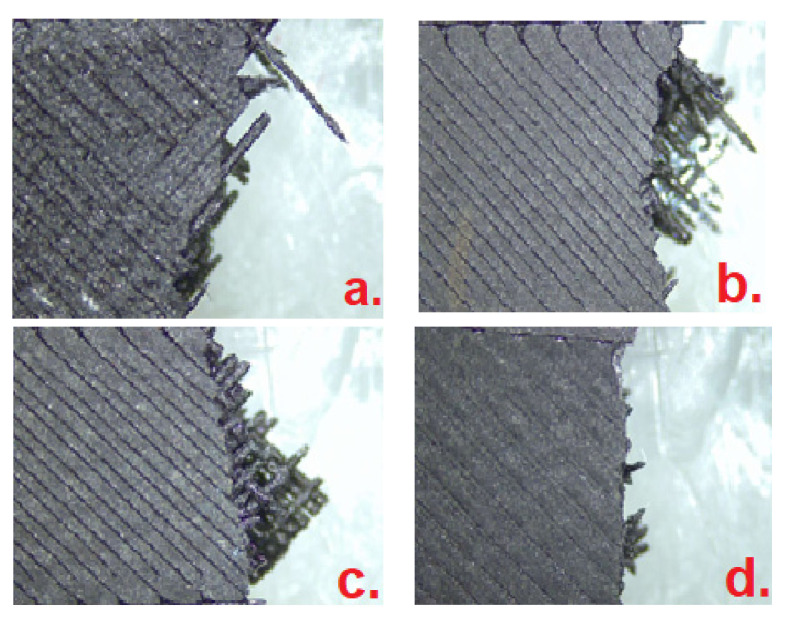
Macroscopic observation after quasi-static tensile test: (**a**) hexagonal, (**b**) triangular, (**c**) rectangular, and (**d**) solid fill.

**Figure 13 polymers-15-00507-f013:**
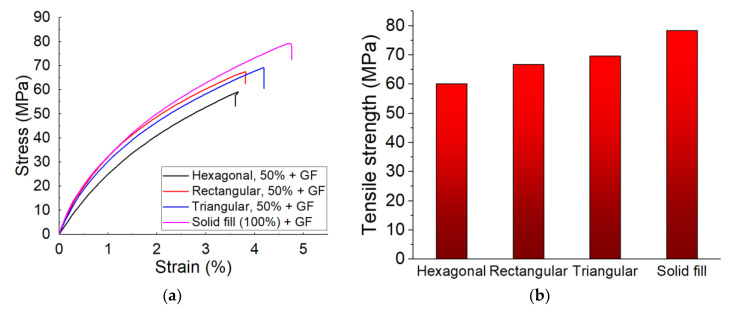
Obtained tensile test curves (**a**) and obtained strength (**b**) of CF-PA6 reinforced with continuous glass fiber specimens [24].

**Figure 14 polymers-15-00507-f014:**
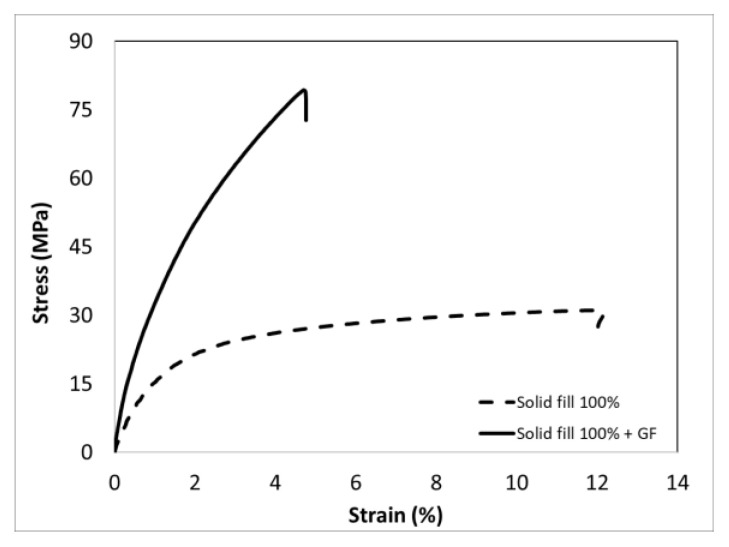
Quasi-static tensile curves of CF-PA6 (Onyx) and CF6-PA6 reinforced with continuous glass fiber (Onyx + GF) (solid fill pattern) [24].

**Figure 15 polymers-15-00507-f015:**
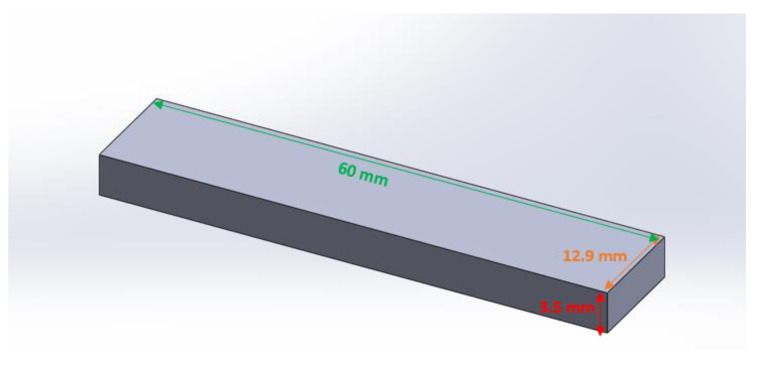
The scheme of the related rectangular composite.

**Figure 16 polymers-15-00507-f016:**
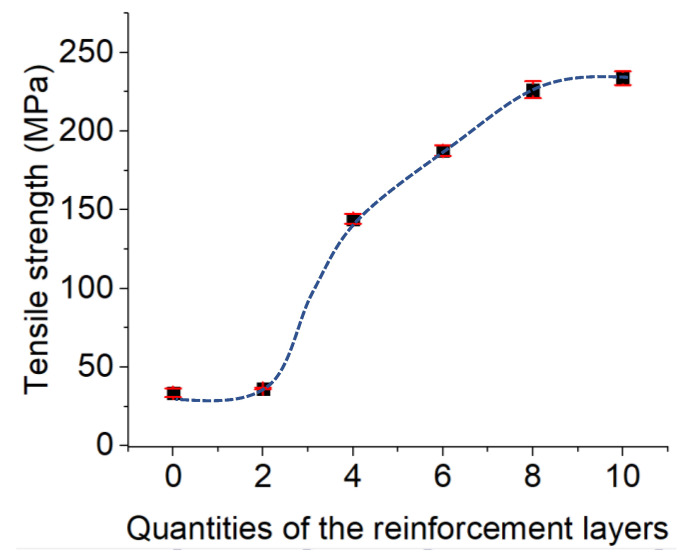
Effect of the density of the continuous reinforcement layers on the tensile strength.

**Figure 17 polymers-15-00507-f017:**
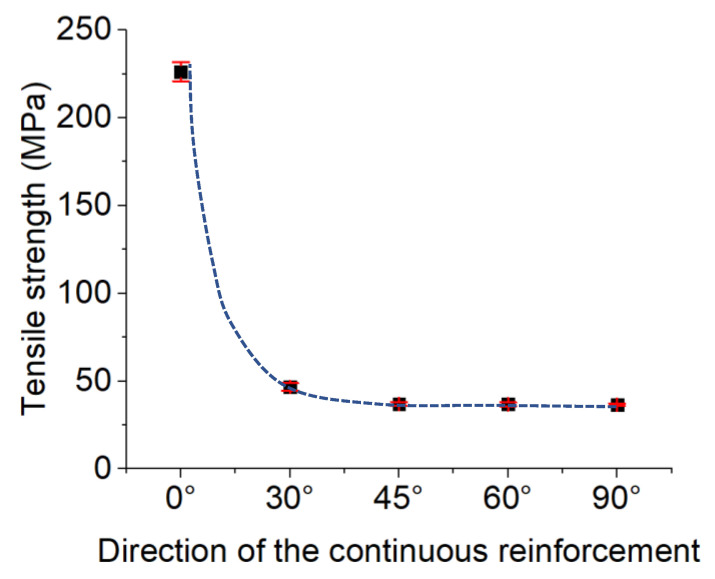
Effect of the direction of the continuous reinforcement on the tensile strength.

**Figure 18 polymers-15-00507-f018:**
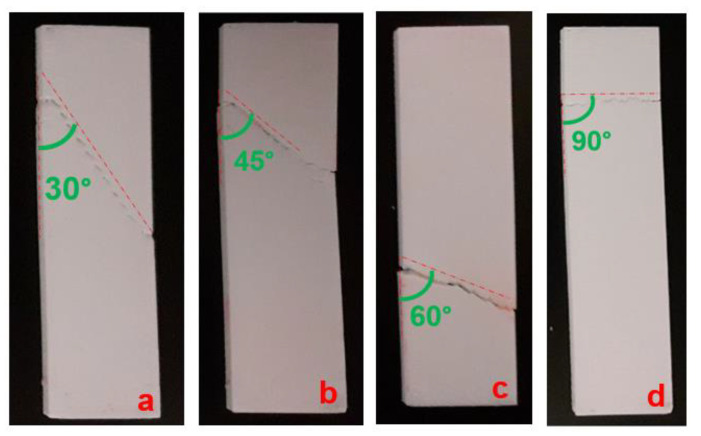
Macroscopic fracture view of the fabricated samples with reinforcement directions of (**a**) 30°, (**b**) 45°, (**c**) 60°, and (**d**) 90°.

**Figure 19 polymers-15-00507-f019:**
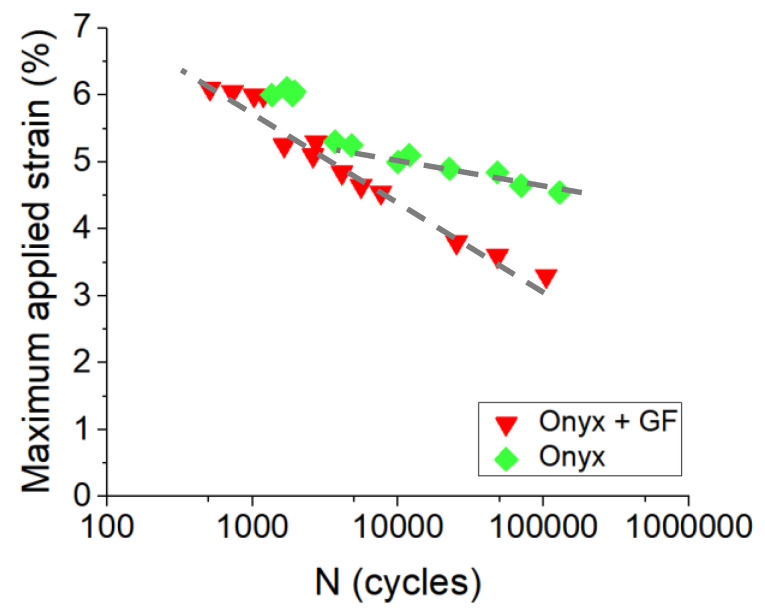
Obtained Wöhler curves concerning CF-PA6 reinforced with continuous glass fiber (Onyx + GF) and CF-PA6 (Onyx) specimens at 10 Hz [24].

**Figure 20 polymers-15-00507-f020:**
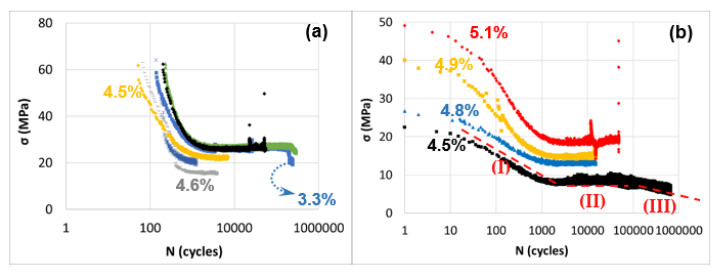
Relative stress (~Young’s modulus) trend in the fatigue test of (**a**) CF-PA6 reinforced with continuous glass fiber (Onyx + GF) specimens and (**b**) CF-PA6 (Onyx) [24].

**Figure 21 polymers-15-00507-f021:**
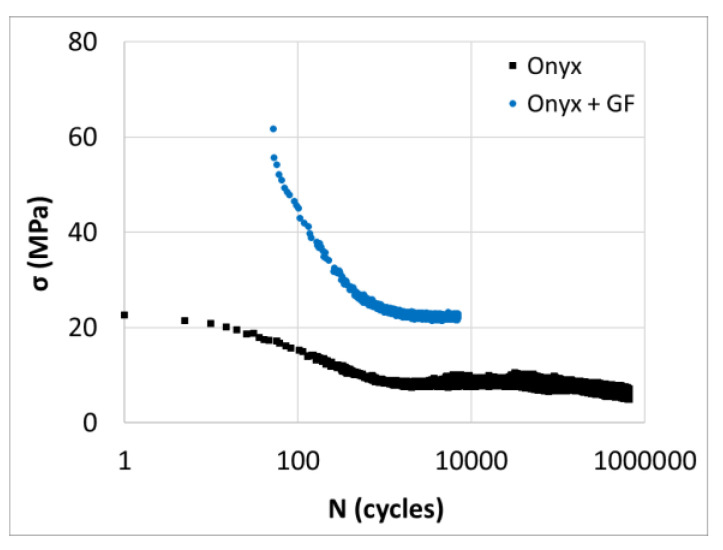
Relative stress (~Young’s modulus) trend within the fatigue test [24].

**Figure 22 polymers-15-00507-f022:**
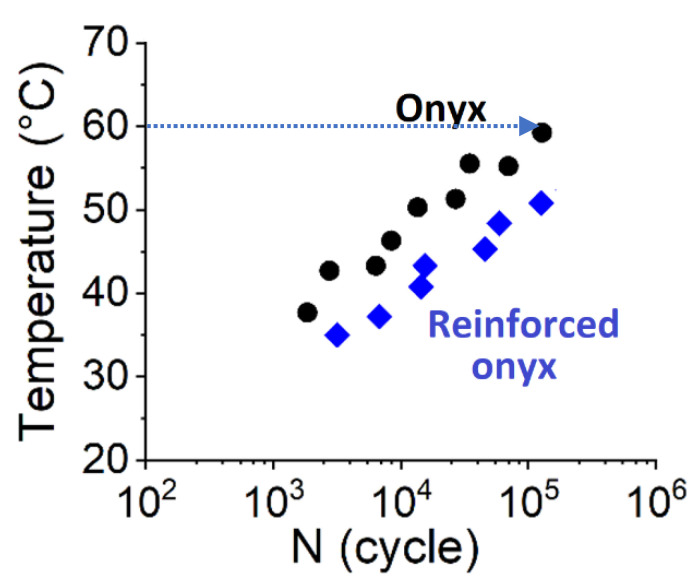
Evolution of the maximum induced temperature during the fatigue test [24].

**Figure 23 polymers-15-00507-f023:**
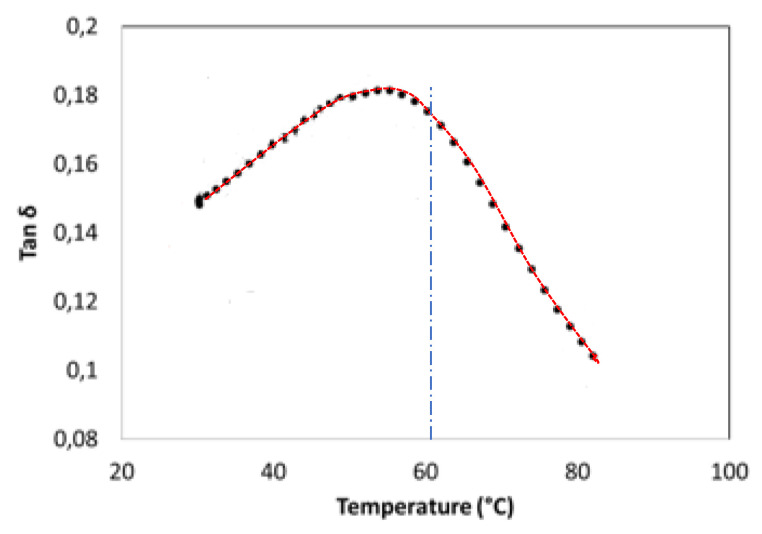
Obtained loss factor evolution versus temperature curve concerning CF-PA6.

**Figure 24 polymers-15-00507-f024:**
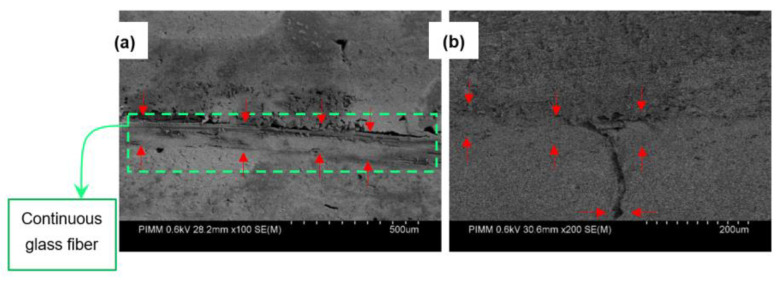
Fractography of the reinforced CF-PA6 with continuous glass fiber (**a**), and CF-PA6 (**b**) specimens [24].

**Table 1 polymers-15-00507-t001:** The characterizations of the utilized raw materials.

	Raw Materials	PA6	CF-PA6
Physical and Chemical Properties			
**Density**		1.1 g/cm^3^	1.2 g/cm^3^
**Glass transition temperature (T_g_)**		45 °C	47 °C
**Crystallization temperature** **(T_c_)**		173 °C	162 °C
**Melting temperature** **(T_m_)**		205 °C	198 °C
**Spool Image**		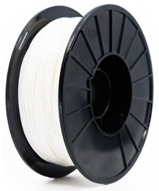	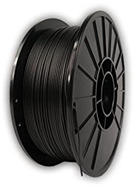

## Data Availability

The data presented in this study are available on request from the corresponding author.

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
