# Peer review of "Thermal, Tensile and Fatigue Behaviors of the PA6, Short Carbon Fiber-Reinforced PA6, and Continuous Glass Fiber-Reinforced PA6 Materials in Fused Filament Fabrication (FFF)"

_polymers, 2023, doi:10.3390/polym15030507_

Round 1

Reviewer 1 Report

The paper presents an interesting approach based on the An Experimental study on the behaviors of the polymer and polymer composite materials in Fused Filament Fabrication (FFF). However, the innovation of the current research work should be further highlighted and emphasized. At the same time, the authors should consider the following comments to greatly improve the quality of the paper.

1. In the abstract, add a final statement that highlights the importance of this research and its possible potentials. 

2. In the keywords list, kindly be more specific in selecting the keywords. Hence, change the two keywords: polymers; composites

3. The introduction needs to be improved by relating to the mechanics of the studied materials and their mechanical characteristics. The references to be included are: 10.1177/0021998318790093, 10.1016/j.polymertesting.2017.09.009.

4. Kindly add a table that describes the main physical and chemical properties of the raw materials used in this study.

5. Were the preparation methods described by the authors come in accordance with a certain standard or do they follow previous procedures?

6. For the Quasi-static tensile tests, what was the reason for the specified test conditions in this research? Do the speed of test value relate to a specific application? Why did the authors choose the only considered strain-ratio to be 0.1?

7. How many samples were used per configuration for the tensile tests? What was the standard used for the test?

8. The two diagrams in Figure 19 aren't clear. The authors need to re-draw then and to explain the exact outcomes from these curves.

9. The conclusions and perspectives need to be modified to summarize the research outcomes in short statements with clear observations.

Author Response

Responses to the reviewers

In what follows, I explain how I revised the submitted version of the paper and I address the issues raised in the reviewer’s comments (recalled in italic bleu characters). It is worth mentioning that changes and modifications, made while revising the paper, are written by red characters in the text of the revised manuscript.

Answer the questions:

Reviewer #1:

Comments to the Author

The paper presents an interesting approach based on the An Experimental study on the behaviors of the polymer and polymer composite materials in Fused Filament Fabrication (FFF). However, the innovation of the current research work should be further highlighted and emphasized. At the same time, the authors should consider the following comments to greatly improve the quality of the paper.

  1. In the abstract, add a final statement that highlights the importance of this research and its possible potentials.

Answer: Thanks for this comment. Changes and modifications have been applied in the manuscript. Therefore, the below statement is added to the revised manuscript in red color.

“Therefore, this study is helpful for the manufacturers and the designers to understand the behaviors of the materials during the FFF process and subsequently the behaviors of the manufactured parts as the function of the different process parameters.”

  1. In the keywords list, kindly be more specific in selecting the keywords. Hence, change the two keywords: polymers; composites

Answer: Thanks for this comment. The required modifications have been applied in the manuscript, which are determined by the red characters in the revised manuscript:

Material extrusion; Rheological behavior; Mechanical properties; Temperature profile

  1. The introduction needs to be improved by relating to the mechanics of the studied materials and their mechanical characteristics. The references to be included are: 10.1177/0021998318790093, 10.1016/j.polymertesting.2017.09.009.

Answer: Thanks for this comment. The required modifications have been applied in the manuscript.

  1. Kindly add a table that describes the main physical and chemical properties of the raw materials used in this study.

Answer: Thanks for this comment. Table 1 (below) is added in the revised manuscript:

Raw materials

PA6

CF-PA6

Physical and chemical properties

Density

1.1 g/cm3

1.2 g/cm3

Glass transition temperature (Tg)

45°C

47°C

Crystallization temperature

(Tc)

173°C

162°C

Melting temperature

(Tm)

205°C

198°C

Spool Image

  1. Were the preparation methods described by the authors come in accordance with a certain standard or do they follow previous procedures?

Answer: Thanks for this comment. The preparation methods are explained in the reference [5] as is cited below the fig. 1. Also, the prepration of the different characterization/ test samples are determined by the red characters in the revised manuscript as below;

As for the investigation of the fiber reinforcement impact and also the process parameters effects on the rheological behavior of the materials during FFF process, a single wall layer specimen (Fig. 1) was designed.

Also, the location of the required specimens for the subsequent characterizations are determined in Fig. 1.  

  1. For the Quasi-static tensile tests, what was the reason for the specified test conditions in this research? Do the speed of test value relate to a specific application? Why did the authors choose the only considered strain-ratio to be 0.1?

Answer: Thanks for this comment. An extruded filament faced with various heat transfer mechanisms due to the different heat sources among the manufacturing process. Consequently, the physical contacts resulting from the deposition mechanisms.

A single deposition road (a single wall layer sample) has been modeled in order to study the impact of short reinforcement and process parameters on the rheological behavior of the materials during FFF process. The selection of the specimens has explained in the previous work as is cited [5] Because the following characteristics were realized by the single wall specimen:

  • Homogeneous deposition of filaments on top of each other
  • Unidirectional deposition of filaments (Consideration the time of deposition for each filament)
  • Same convection of layers with the environment
  • Same conduction between layers
  • 1st layer: conduction with support (and with 2nd layer) simultaneously (for thermal characterization)
  • 2nd, 3rd, …, nth layers: same conduction with each other’s
  • Symmetric effect of environment and platform temperature on the solidification of material while cooling down.

The test speed has been selected based on the previous study [5] and also literature review.

As for the composite materials, the stain-ratio value of 0.1 was found to be so common, based on the performed literature review.

  1. How many samples were used per configuration for the tensile tests? What was the standard used for the test?

Answer: Thanks for this comment. Minimum three specimens were prepared for tensile test. Therefore, the below text is determined by the red characters in the revised manuscript as below:

A minimum of three specimens were prepared for each condition in order to conduct the quasi-static tensile tests.

Also, as is explained in the manuscript the tensile test specimens concerning the study of the impact of the short reinforcement and process parameters on the rheological behavior of the materials during FFF process were according ISO 527-2. Also, the tensile test specimens concerning the rheological behavior of the additive manufactured specimens after the production process were according ISO 527-1.

  1. The two diagrams in Figure 19 aren't clear. The authors need to re-draw then and to explain the exact outcomes from these curves.

Answer: Thanks for this comment. The required modifications have been applied in the manuscript.

  1. The conclusions and perspectives need to be modified to summarize the research outcomes in short statements with clear observations.

Answer: Thanks for this comment. Changes and modifications have been applied in the manuscript. Therefore, the below statement is added to the revised manuscript in red color.

The primary purpose of this research was to study the rheological characteristics of the materials during and after the FFF process. As a result of this study, we can gain a better understanding of the behaviors of polymers and polymer composite materials, as well as how reinforcements function during and after the FFF process. The obtained results will enable us to improve the FFF processed parts. In fact, the fundamental problem of the produced parts by the FFF process is the presence of the porosities/voids inside the manufactured parts and the subsequent poor adhesion between the deposited layers. In this study, influences of the various process parameters on the material behaviors during and after FFF process were investigated. The physicochemical, thermal, and mechanical assessment exhibited that the crystallinity degree was influenced by the process parameters modification. The crystallinity degree modifications could affect the material diffusion during the cooling stage and the subsequent bonding of the deposited adjacent filaments. The strength of this bonding directly affects the mechanical behaviors of the manufactured specimens. Therefore, a precise and local measurement of the temperature on the scale of the diameter of the filaments during FFF process was performed. Also, by comparing the obtained temperature evolution curves from PA6 and CF-PA6 specimens it was found out that the existence of the chopped carbon fibers could affect the temperature evolution curves during the FFF process.

The obtained results from three points bending fatigue and tensile behavior of the manufactured composite materials reveals:

  • The FFF processed CF-PA6 specimens with solid fill patterns had superior mechanical properties and stiffness under tension.

  • Based on the study of the effect of the continuous glass fiber reinforcements on the fatigue behavior of the manufactured specimens, a reduction in fatigue life was observed in the manufactured CF-PA6 and continuous glass fiber reinforced CF-PA6 specimens due to an increase in the induced temperature. By considering this induced temperature, known as self-heating, we could determine how a polymer behaves viscously, especially at the glass transition zone. It was observed that CF-PA6 reinforced with continuous glass fiber specimens exhibited a lower level of self- heating compared to CF-PA6 specimens without continuous glass fiber reinforcement during the fatigue test. based on the microscopic analysis of the damage mechanisms during the per-formed fatigue tests, the fracture surface of the continuous glass fiber rein-forced CF-PA6 specimens was characterized by continuous glass fiber pull-out and subsequent debonding of the deposited layers.

  • As for the study of the density of continuous glass fiber in FFF process of polymer-based composites, by increasing the density of the reinforcement layers the strength of the manufactured specimens were increased until a specific density value then became stable.

  • As for study the effect of the direction of the continuous reinforcement in the FFF processed polymer composite materials, the five directions of 0°, 30°, 45°, 60°, and 90° were investigated. The highest strength was related to the manufactured parts, in which the reinforcements were in the applied stress direction (0°). As the direction of the utilized reinforcement deviated more from the applied stress direction, the strength of the manufactured parts was de-creased. The fracture of the tensile tested samples occurred in the direction of the deposited continuous reinforcement.

In the next study the effect of other continuous reinforcement such as carbon fiber and Kevlar fiber on the tensile and the fatigue behavior of the printed specimens will be studied. Therefore, their effect will be compared with each other.

Reviewer 2 Report

The paper presents multiple analyses performed on FFF 3D printed specimens, using polyamide, carbon fiber reinforced polyamide and glass fiber reinforced polyamide.

The paper requires restructuring. The current form is ambiguous. I suggest the following to the authors:

-clear formulation of the title, showing the materials tested and what behaviour was observed. The current title is too general;

- the abstract is inadequate. It is rather a part of the introduction. The abstract does not clearly define the purpose of the work.

- The introduction is general and uses general quotations. The introduction should contain several references related to the purpose of the paper and it should be detailed what is concluded in these references.

-notes and symbols should be clearly defined and kept consistent throughout the paper (e.g. different symbols for layer thickness are used in Figure 8).

-In figure 2 two materials are shown. In the previous paragraphs, 3 materials are mentioned.

-Section 2 Materials and methods must be clearly structured. The authors only present a list of applied methods. The test structure and conditions, the  shape, geometry and number of specimens should be described. It is not clear when tensile specimens according to ISO 527-1 were used and when specimens cut from single wall parts were used.

-Please describe how stability was ensured in single wall printing. How were specimens cut out in single wall prints (for mechanical tests)?

-Explain why two printers were used.  

-Explain why some tests are done on PA and CF-PA6 and not on all 3 types of materials. 

- when varying the working parameters it is necessary to show the values for the fixed parameters.

- Figure 8 has two parts. It is necessary to detail them.

- The presentation in one paper of the influence of a large number of parameters makes the work difficult to analyse. I consider that the research results can be published in two separate papers

Author Response

Responses to the reviewers

In what follows, I explain how I revised the submitted version of the paper and I address the issues raised in the reviewer’s comments (recalled in italic bleu characters). It is worth mentioning that changes and modifications, made while revising the paper, are written by red characters in the text of the revised manuscript.

Answer the questions:

Reviewer #2:

Comments to the Author

The paper presents multiple analyses performed on FFF 3D printed specimens, using polyamide, carbon fiber reinforced polyamide and glass fiber reinforced polyamide.

The paper requires restructuring. The current form is ambiguous. I suggest the following to the authors:

-clear formulation of the title, showing the materials tested and what behaviour was observed. The current title is too general;

Answer: Thanks for this comment. The modifications have been applied in the title of the revised manuscript. Therefore, the below statement is added to the revised manuscript in red color.

Study the thermal, tensile and fatigue behaviors of the PA6, short carbon fiber-reinforced PA6, and continuous glass fiber-reinforced PA6 materials in Fused Filament Fabrication (FFF).

- the abstract is inadequate. It is rather a part of the introduction. The abstract does not clearly define the purpose of the work.

Answer: Thanks for this comment. The required modifications have been applied in the abstract section of the revised manuscript. Therefore, the modified abstract is:

Utilization of additive manufacturing (AM) is widespread in the industries due to its unique capabilities. The material extrusion methods have been developed extensively for manufacturing the polymer and polymer composite materials. The raw material in filament form will be liquefied in the liquefier section and consequently will be extruded and deposited on the bed platform. The designed parts will be manufactured layer by layer. Therefore, there is a gradient of temperature due to the existence of the cyclic reheating related to each deposited layer by the newer deposited ones. So, the stated temperature evolution will have a significant role on the rheological behavior of the materials during this manufacturing process. Furthermore, each process parameters can affect this cyclic temperature profile. In this study, different process parameters concerning the manufacturing process of polymer and polymer composite samples have been evaluated according to their cyclic temperature profiles. In addition, the manufactured parts by the additive manufacturing process (the extrusion method) can behave different comparing to the manufactured parts by the conventional methods. Accordingly, it is attempted to experimentally investigate the rheological behavior of the manufactured parts after the manufacturing process. Thus the three-point bending fatigue and the tensile behavior of the manufactured samples were studied. Accordingly, the effect of the reinforcement existence, its direction and density on the tensile behavior of the manufactured samples were studied. Therefore, this study is helpful for the manufacturers and the designers to understand the behaviors of the materials during the FFF process and subsequently the behaviors of the manufactured parts as the function of the different process parameters.

- The introduction is general and uses general quotations. The introduction should contain several references related to the purpose of the paper and it should be detailed what is concluded in these references.

Answer: Thanks for your valuable comment. Kindly it was attempted to enrich the introduction by the added the references in the revised manuscript in red color.

An investigation was conducted by Ding et al. [21] to determine the influence of liquefier temperature on the mechanical properties of the FFF processed PEEK and PEI. They discovered that the flexural strength was gradually improved as the temperature increased. Berretta et al. [22] manufactured the reinforced PEEK with 1% and 5% car-bon nanotubes (CNTs). They reported that the CNTs didn’t have significant effect on the mechanical behaviors of the PEEK processed specimens. They introduced the noz-zle temperature as one of the most crucial parameters in the FFF process, due to its di-rect contact with the polymer. Yang et al. [23] studied the effect of the thermal pro-cessing condition on mechanical behaviors and crystallinity of the PEEK material. Based on the related results, crystallinity increased from 17% to 31% in response to the increase in ambient temperature from 25 °C to 200 °C.

-notes and symbols should be clearly defined and kept consistent throughout the paper (e.g. different symbols for layer thickness are used in Figure 8).

Answer: I appreciate your helpful comment. The related figure (Fig. 8) is modified in the revised manuscript.

(b)

(a)

(Tc)

-In figure 2 two materials are shown. In the previous paragraphs, 3 materials are mentioned.

Answer: Your comment is greatly appreciated. The “materials” section of the manuscript is revised in red color.

2.1. Materials

The selected materials for this study were polyamide 6 (PA6) and short carbon fiber-reinforced polyamide 6 (Onyx or CF-PA6) produced by MarkForged®. The chopped carbon fibers had a mass content of 6.5% in the CF-PA6 filament based on the pyrolysis process. The characteristics of the utilized filaments as the raw materials are presented below (Table 1):

Raw materials

Physical and chemical properties

PA6

CF-PA6

Density

1.1 g/cm3

1.2 g/cm3

Glass transition temperature (Tg)

45°C

47°C

Crystallization temperature

(Tc)

173°C

162°C

Melting temperature

(Tm)

205°C

198°C

Spool Image

Table 1. The characterizations of the utilized raw materials

As for the investigation of the fiber reinforcement impact and also the process parameters effects on the rheological behavior of the materials during FFF process, a single wall layer specimen (Fig. 1) was designed. This specimen let us study the effect of the selected process parameters on the adhesion and the bonding of the deposited layers. Also the location of the required specimens for the subsequent characterizations are determined in Fig. 1. Two different printers have been utilized during our studies. To study the effects of the process parameters, Flashforge ADVENTURER-3 was utilized. Moreover, as for studying the infill percentage, infill pattern effects, and fatigue behaviors, Markforged-Mark Two printer was utilized.

-Section 2 Materials and methods must be clearly structured. The authors only present a list of applied methods. The test structure and conditions, the shape, geometry and number of specimens should be described. It is not clear when tensile specimens according to ISO 527-1 were used and when specimens cut from single wall parts were used.

Answer: I appreciate your helpful comment. Kindly it was attempted to clarify the test methods. So, the below modifications were applied in the revised manuscript in red color.

As was stated, two different conditions for studying the tensile behavior were conducted. The first condition was considered to study the effect of the process parameters on the manufacturing of the polymer and composite specimens. The second condition was applied to study the impact of fill percentage of the polymer and the different determined densities and directions of the utilized continuous reinforcements. Therefore, based on ISO 527-2, tensile test specimens were sliced/cut from the printed single wall layer sample (Fig. 1), in order to study the impact of short reinforcement and process parameters on the rheological behavior of the materials during FFF process. The required tensile specimens were cut from the printed single-wall samples utilizing a tensile sample-cutting die and applying the homogenous force. Also, the homogeneity of the printed single-wall layers was ensured by using the caliper. Moreover, in order to investigate the rheological behavior of the additive manufactured specimens after the production process, the related tensile test specimens were printed based on the standard ISO 527-1. The quasi-static tensile experiment was conducted by the means of INSTRON 5966 machine with the displacement rate of 5 mm/min and the loading cell of 10 kN. A minimum of three specimens were prepared for each condition in order to conduct the quasi-static tensile tests.

In order to perform a fatigue test, at least three specimens were prepared for each condition.

-Please describe how stability was ensured in single wall printing. How were specimens cut out in single wall prints (for mechanical tests)?

Answer: Thanks for your valuable comment. Kindly it was attempted to clarify the test methods. So, the below modifications were applied in the revised manuscript in red color.

The required tensile specimens were cut from the printed single-wall samples utilizing a tensile sample-cutting die and applying the homogenous force. Also, the homogeneity of the printed single-wall layers was ensured by using the caliper.

-Explain why two printers were used. 

Answer: My sincere thanks go out to you for your valuable comment. The below modification has been applied in the revised manuscript in red color.

This is because Markforged-Mark Two printers can provide the possibility of manufacturing composite objects with continuous reinforcement.

-Explain why some tests are done on PA and CF-PA6 and not on all 3 types of materials.

Answer: Your priceless comment is greatly appreciated.

Two main spools of filament were utilized as the main raw materials, which were PA6 and CF-PA6 (Onyx). The related conducted study on the effect of the process parameters and the thermal behavior was applied to PA6 and CF-PA6. This lets us study the effect of short fiber reinforcements (carbon fibers) on the rheological behavior of the materials during the manufacturing process.

On the other hand, the mechanical behavior of the manufactured objects (after the manufacturing process). In this section, the continuous reinforcements composites (reinforced by the continuous glass fibers) were studied, too. This lets us study the effect of continuous reinforcements on mechanical behavior and also the self-heating phenomena during fatigue loading. The effect of continuous reinforcements on the rheological behavior of the materials during the manufacturing process will be studied in the next research project.

- when varying the working parameters it is necessary to show the values for the fixed parameters.

Answer: I greatly appreciate your incredibly valuable comment. The below clarification is determined in the revised manuscript in red color.

The selected process parameters were print speed (13 mm/s, 15 mm/s, and 17 mm/s), liquefier temperature (220°C, 230°C, and 240°C), layer height (0.1 mm, 0.2 mm, and 0.3 mm), and bed platform temperature (25°C, 50°C, 60°C, and 80°C). The aforementioned process parameters were studied individually by considering the rest of the process parameters as constant.

- Figure 8 has two parts. It is necessary to detail them.

Answer: I greatly appreciate your incredibly valuable comment. The caption of this figure is modified in the revised manuscript in red color.

(b)

(a)

(Tc)

Figure 7. Monitoring of the temperature rise during the additive manufacturing process of CF-PA6 specimens with the different selected layer height values; (a) upper limits and (b) lower limits.

- The presentation in one paper of the influence of a large number of parameters makes the work difficult to analyse. I consider that the research results can be published in two separate papers

Answer: I greatly appreciate your incredibly valuable comment. Indeed, as one of the main aim of this manuscript, it was tried to study the behaviors of polymer and composite materials during and also after the manufacturing process. The lack of this approach of the study in the field of FFF process was sensed significantly. But, the next research will be published in the separated papers to facilitate the analysis affairs.

Reviewer 3 Report

1. Page #8 (line 247-249): "The stated two phenomena were fluidity decrease of the printed materials and residual/retained temperature increase of the deposited layers during the cyclic raised temperature during FFF process."

- The message not clear. Please elaborate.

2. Page #9 (line 289): " Liquefier temperature of 80°C".

- It should be be the print bed temperature?

3. Page #10 (line 296): "3.2 Study the rheological behavior of polymer and polymer composite material after FFF process:"

- Please show pictures of printed physical parts for better understanding

4. Page #11 (line 336): "3.2.1.2. Effect of the continuous reinforcement on different fill patterns"

- Please explain why continuous reinforcement of GF was chosen? Also, show printing process setup and printed physical parts for better understanding.

5. Page #12 (line 360-361): "a rectangular sample with thickness of 3.5 mm was designed as the tensile test specimens."

- show picture of  the specimen geometry/ dimensions

6. Page #15 (fig. 17):

- Poor quality of the figure. Labels a, b, c, d are not visible; crack not visible; red dotted lines can be shown parallel but a little away from the crack (not on top of it).

7. Page #16 (fig. 19):

- Provide legends to define what each curve indicates.

8. Page #17 (fig. 22):

- Provide legends to define what each curve indicates.

9. Page #17 (line 488-492): "Indeed, considering the induced temperature during the fatigue test is so vital. In such a way that the maximum induced temperature during the fatigue cycles is a crucial point for considering to prognosticate the fatigue behavior of the printed polymer and continuous fiber reinforced polymer composite materials."

- The message not clear. Please elaborate.

10. Conclusions:

- Please provide a few comments on future scope of the work.

Author Response

Responses to the reviewers

In what follows, I explain how I revised the submitted version of the paper and I address the issues raised in the reviewer’s comments (recalled in italic bleu characters). It is worth mentioning that changes and modifications, made while revising the paper, are written by red characters in the text of the revised manuscript.

Answer the questions:

Reviewer #3:

Comments to the Author

  1. Page #8 (line 247-249): "The stated two phenomena were fluidity decrease of the printed materials and residual/retained temperature increase of the deposited layers during the cyclic raised temperature during FFF process."

- The message not clear. Please elaborate.

Answer: Thanks for this comment. The more clarification has been applied in the revised manuscript. Therefore, the below stated-modification is determined in the revised manuscript by red color:

The stated two phenomena were firstly the fluidity decrease of the printed materials and secondly the residual/retained temperature increase of the deposited layers during the cyclic raised temperature during FFF process. The peaks (upper limits) of the obtained time-temperature curves during the manufacturing process revealed the first-stated phenomenon, also the lower limits of these curves revealed the second-stated phenomenon.

  1. Page #9 (line 289): " Liquefier temperature of 80°C".

- It should be the print bed temperature?

Answer: I greatly appreciate your incredibly valuable comment. The below clarification is determined in the revised manuscript in red color:

« the bed temperature of 80°C »

  1. Page #10 (line 296): "3.2 Study the rheological behavior of polymer and polymer composite material after FFF process:"

- Please show pictures of printed physical parts for better understanding

Answer: Thanks for your valuable comment. The related photos have been added to the revised manuscript:

Continuous glass fibers

CF-PA6

  1. Page #11 (line 336): "3.2.1.2. Effect of the continuous reinforcement on different fill patterns"

- Please explain why continuous reinforcement of GF was chosen? Also, show printing process setup and printed physical parts for better understanding.

Answer: Thanks for your comment.

In this study continuous reinforcement of GF was selected to be utilized. But in the next study other reinforcement will be selected and applied.

As for the printed physical parts, the related photo is added (Figure 10), which is in the reply of the previous comment, too.

  1. Page #12 (line 360-361): "a rectangular sample with thickness of 3.5 mm was designed as the tensile test specimens."

- show picture of the specimen geometry/ dimensions

Answer: Thanks for your comment. The scheme of the related rectangular-shaped specimens is determined per below:

Also figure 18 depicts the real pictures of the related specimens.

  1. Page #15 (fig. 17):

- Poor quality of the figure. Labels a, b, c, d are not visible; crack not visible; red dotted lines can be shown parallel but a little away from the crack (not on top of it).

Answer: Thanks for this comment. Fig. 18 is modified in the revised manuscript:

  1. Page #16 (fig. 19):

- Provide legends to define what each curve indicates.

Answer: Thanks for this comment. The required modifications have been applied in the manuscript.

  1. Page #17 (fig. 22):

- Provide legends to define what each curve indicates.

Answer: Thanks for this comment. In fact, this curve (there is just one curve in this figure) is concerning the loss factor evolution versus temperature related to CF-PA6. This curve is obtained from DMTA characterization. This curve revealed the glass transition zone of the polymer matrix. Then introducing of this cure is determined in red color in the revised manuscript:

Consequently, the stiffness of the matrix agent (polymer) was decreased based on the obtained loss factor evolution versus temperature curve from the performed DMTA test Fig. 23.

  1. Page #17 (line 488-492): "Indeed, considering the induced temperature during the fatigue test is so vital. In such a way that the maximum induced temperature during the fatigue cycles is a crucial point for considering to prognosticate the fatigue behavior of the printed polymer and continuous fiber reinforced polymer composite materials."

- The message not clear. Please elaborate.

Answer: I greatly appreciate your incredibly valuable comment. The related text is modified in the revised manuscript in red color:

In fact, it is crucial to take into account the induced temperature during the fatigue test. Therefore, a critical point for predicting the fatigue behavior of the printed polymer and continuous fiber reinforced polymer materials is the maximum induced temperature during the fatigue cycles.

  1. Conclusions:

- Please provide a few comments on future scope of the work.

Answer: Thanks for this valuable comment. The below text is added in the revised manuscript in red color:

In the next study the effect of other continuous reinforcement such as carbon fiber and Kevlar fiber on the tensile and the fatigue behavior of the printed specimens will be studied. Therefore, their effect will be compared with each other.

Round 2

Reviewer 1 Report

The paper can be accepted.

Author Response

Responses to the reviewers

In what follows, I explain how I revised the submitted version of the paper and I address the issues raised in the reviewer’s comments (recalled in italic bleu characters). It is worth mentioning that changes and modifications, made while revising the paper, are written by red characters in the text of the revised manuscript.

Answer the questions:

Reviewer #1:

Comments and Suggestions for Authors

The paper can be accepted.

Answer: I greatly appreciate your kind consideration.

Reviewer 2 Report

The authors have restructured the paper but further clarification is needed on the methodology applied:

- The following paragraph is in contradiction with Figures 10 and 12:

"As was stated, two different conditions for studying the tensile behavior were conducted. The first condition was considered to study the effect of the process parameters on the manufacturing of the polymer and composite specimens. The second condition was applied to study the impact of fill percentage of the polymer and the different determined densities and directions of the utilized continuous reinforcements. Therefore, based on 139 ISO 527-2, tensile test specimens were sliced/cut from the printed single wall layer sample (Fig. 1), in order to study the impact of short reinforcement and process parameters on the rheological behavior of the materials during FFF process. The required tensile specimens were cut from the printed single-wall samples utilizing a tensile sample-cutting die and applying the homogenous force."

- It is not explained how the stability of the material was ensured for the single wall sample printing; I recommend inserting images to clarify this process.

Author Response

Responses to the reviewers

In what follows, I explain how I revised the submitted version of the paper and I address the issues raised in the reviewer’s comments (recalled in italic bleu characters). It is worth mentioning that changes and modifications, made while revising the paper, are written by red characters in the text of the revised manuscript.

Answer the questions:

Reviewer #2:

Comments to the Author

The authors have restructured the paper but further clarification is needed on the methodology applied:

- The following paragraph is in contradiction with Figures 10 and 12:

"As was stated, two different conditions for studying the tensile behavior were conducted. The first condition was considered to study the effect of the process parameters on the manufacturing of the polymer and composite specimens. The second condition was applied to study the impact of fill percentage of the polymer and the different determined densities and directions of the utilized continuous reinforcements. Therefore, based on 139 ISO 527-2, tensile test specimens were sliced/cut from the printed single wall layer sample (Fig. 1), in order to study the impact of short reinforcement and process parameters on the rheological behavior of the materials during FFF process. The required tensile specimens were cut from the printed single-wall samples utilizing a tensile sample-cutting die and applying the homogenous force."

Answer: I greatly appreciate your incredibly valuable comment. For the better clarification, the above text and also figure 1 are revised and clarified in the revised manuscript in red color as below. Kindly it is remarked that figure 1 is concerning the considered “first condition” and figures 10 and 12 are concerning the considered “second condition”.

As was stated, two different conditions for studying the tensile behavior were conducted. The first condition was considered to study the effect of the process parameters on the manufacturing of the polymer and composite specimens (fig. 1). Therefore, based on ISO 527-2, tensile test specimens were sliced/cut from the printed single-wall layer sample (fig. 1), in order to conduct the related study of the first condition. The required tensile specimens were cut from the printed single-wall samples utilizing a tensile sample-cutting die and applying the homogenous force. Also, the homogeneity of the printed single-wall layers was ensured by using the caliper. The second condition was applied to study the impact of fill percentage of the polymer and the different determined densities and directions of the utilized continuous reinforcements.

It is not explained how the stability of the material was ensured for the single wall sample printing; I recommend inserting images to clarify this process.

Answer: Your comment is incredibly valuable and I greatly appreciate it. In fact, the thickness of the printed samples was measured and inspected to be ensured the stability of the related single-wall samples as is stated in the revised manuscript in red color as below:

Also, the homogeneity of the printed single-wall layers was ensured by using the caliper.

In addition, the process of the preparation of the related tensile test specimens is inserted in Figure 1.

Figure 1. The printed single-wall specimens and preparation of the tensile test specimens [5]

Round 3

Reviewer 2 Report

The process of printing a single-walled specimen is not clearly presented. It is not clear how the stability of the material is ensured in this printing (no collapse or buckling?). At the same time the images inserted in Figure 1 show a poor quality of the cut-out outline. How does the quality of the outline of the specimen influence the results?

Author Response

Answer: Thanks for your comment.

It is remarked that as the material deposited became solid completely. So, no collapse was occurred during the printing process with the selected process parameters.

As is stated “the required tensile specimens were cut from the printed single-wall samples utilizing a tensile sample-cutting die and applying the homogenous force. Also, the homogeneity of the printed single-wall layers was ensured by using the caliper for thickness measurement”. Following the cutting of the tensile test specimens from the single-wall layers, a caliper was used to ensure gauge length dimensions were uniform. In addition, the observation of the specimens under the optical microscope (OM) as for controlling the quality of the specimens after manufacturing and cutting process was taken into account. So, by considering these policies tried to stability of the related single-walled to be controlled. No collapse or buckling were occurred. Also, a better image is inserted in Fig. 1.

Therefore, the below text is added to the revised manuscript in the red color.

Also, the homogeneity of the printed single-wall layers was ensured by using the caliper for thickness measurement. Following the cutting of the tensile test specimens from the single-wall layers, a caliper was used to ensure gauge length dimensions were uniform. In addition, the observation of the samples under the optical microscope (OM) to control the quality of the specimens after the manufacturing and cutting process was taken into account.

Fig. 1. The printed single-wall specimens and preparation of the tensile test specimens [5]

Round 4

Reviewer 2 Report

Accept in present form.